# EPHA3 Could Be a Novel Prognosis Biomarker and Correlates with Immune Infiltrates in Bladder Cancer

**DOI:** 10.3390/cancers15030621

**Published:** 2023-01-19

**Authors:** Junpeng Liu, Zewen Zhou, Yifan Jiang, Yuzhao Lin, Yunzhi Yang, Chongjiang Tian, Jinwen Liu, Hao Lin, Bin Huang

**Affiliations:** 1Department of Urology, The Second Affiliated Hospital of Shantou University, Medical College, Shantou 515041, China; 2Department of Urology, The Fifth Affiliated Hospital of Guangzhou Medical University, Guangzhou 510700, China; 3Department of Urology, The First Affiliated Hospital of Sun Yat-Sen University, Guangzhou 510080, China

**Keywords:** EPH receptor A3, bladder cancer, TCGA, ROC, GEO, overall survival, biomarker, immune-infiltrating cells, immune microenvironment

## Abstract

**Simple Summary:**

Bladder cancer is a type of malignant tumor. Eighty percent of patients are diagnosed with non-muscle-invasive bladder cancer (NMIBC), which has a good prognosis, but it reoccurs often. The other patients are diagnosed with muscle-invasive bladder cancer (MIBC), which is associated with high mortality and metastasis rates. We urgently need to explore novel early diagnostic biomarkers and therapeutic targets. Our study reveals the anti-cancer role of EPHA3 and its potential value in immunotherapy.

**Abstract:**

Purpose: To assess the mechanism of EPH receptor A3 (EPHA3) and its potential value for immunotherapy in BLCA. Materials and Methods: The Cancer Genome Atlas (TCGA) bladder cancer (BLCA) database and the Gene Expression Omnibus (GEO) database were used for assessing whether EHPA3 could be used to predict BLCA prognosis. This work carried out in vitro and in vivo assays for exploring how EPHA3 affected the biological behaviors. The downstream pathway was explored using a Western blotting technique. The CIBERSORT, ESTIMATE, TIMER, and TIDE tools were used to predict the immunotherapy value of EPHA3 in BLCA. Results: EPHA3 was poorly expressed in BLCA (*p* < 0.05), its high expression is related to a good survival prognosis (*p* = 0.027 and *p* = 0.0275), and it has a good predictive ability for the histologic grade and status of BLCA (area under curve = 0.787 and 0.904). Overexpressed EPHA3 could inhibit BLCA cell biological behaviors, and it be associated with the downregulation of the Ras/*p*ERK1/2 pathway. EPHA3 was correlated with several immune-infiltrating cells and the corresponding marker genes. Conclusions: EPHA3 could be regarded as an acceptable anti-cancer biomarker in BLCA. EPHA3 plays an inhibiting role in BLCA, and it could be the candidate immunotherapeutic target for BLCA.

## 1. Introduction

Bladder cancer (BLCA) refers to the fifth most predominant malignancy, with about 77,000 new incident cases and 16,400 mortalities occurring annually in the USA [1]. BLCA is mainly stratified into two kinds (containing muscle-invasive bladder cancer (MIBC), as well as non-muscle-invasive bladder cancer (NMIBC)) [2]. More than 70% of NMIBC patients have good 5 year overall survival (OS) rates of up to 69%–96% [3]. However, NMIBC may malignantly progress to MIBC [4]. Due to more than one-half of patients having recurrences following radical surgery and developing metastases, the survival outcomes of MIBC patients are unfavorable, with a low 5 year OS rate (48%) among the treated cases [5,6,7]. To improve the therapeutic effect and reduce the rate of poor prognoses, diagnosing patients suffering from BLCA in the early stages is pivotal. However, reliable tumor markers or prognostication models that can accurately predict the occurrence and progression of BLCA are still lacking clinical applications. Hence, there is an urgent need to explore new diagnoses and prognosis markers.

Receptor tyrosine kinases (RTKs) are relevant to the initiation, progression, and development of cancers [8,9]. Additionally, Ephrin (EPH) receptor is the largest subfamily among the RTKs [10]. In line with the similarity of the EPH receptors and their affinity with ligands, EPH receptors can be classified into A and B ones, including nine A receptors and five B receptors. In 1992, Boyd et al. [11] described a new protein that may be associated with tumors from hematopoietic system malignant tumor cell lines; the new protein was an RTK that was designated as EPHA3. It was further shown that the mutation and abnormal expression level of EPHA3 were correlated with lung cancer and gastric cancer, respectively, which confirmed that EPHA3 could be considered as a target in cancer therapy [12,13]. Meanwhile, according to high-throughput screening, EPHA3, as a type of transmembrane receptor, was identified as a binding partner for programmed cell death-ligand 1 (PD-L1), and EphA3/PD-L1 co-expression was associated with a CD8+ effector cell signature [14]. This result revealed the potential of targeting EPHA3 to help regulate the immune-infiltrating microenvironment, which may provide a new tactic in immunotherapy. Nevertheless, the relevance of EPHA3 with a progression to bladder cancer and the patient’s clinicopathological characteristics have not been reported, which deserve further exploration.

The Ras/ERK pathway is highly active in malignant tumors and transmits signals from the receptor of the cell surface to regulate the cellular physiological behavior [15]. There is evidence that Ras/ERK pathway activation is related to regulating BLCA progression [16]. The Ras mutation (mainly in H-Ras) has been observed in 40% of BLCA cases, and its high expression level may lead to tumor formation. This may be associated with the reduced expression of p21, depending on the MAPK signaling pathway. Meanwhile, the negative correlation between p21 and pERK1/2 was observed in human BLCA. All of these results indicate that the Ras/ERK signaling pathway activation may drive BLCA in vivo.

In recent years, immunotherapy has been gradually adopted for the treatment of BLCA [17]. For instance, immune checkpoint inhibitors (ICIs) are gradually applied in advanced BLCA, thus improving the survival rate in some patients [18]. Although the results of different types of trails show promising improvements, only some patients can benefit from immunotherapy [19]. Therefore, it is promising to explore effective immunotherapeutic-related biomarkers in BLCA.

Based on the above-mentioned information, our study aimed to identify the molecular mechanism of EPHA3 in BLCA.

## 2. Material and Methods

### 2.1. Data Source

The Cancer Genome Atlas (TCGA) (https://portal.gdc.cancer.gov/, accessed on 10 November 2022) and GEO databases (http://www.ncbi.nlm.nih.gov/geo/, accessed on 11 November 2022) provided the EPHA3 mRNA expression datasets and related clinical pathologic data, respectively. This work utilized the TIMER 2.0 website (http://timer.comp-genomics.org/, accessed on 10 November 2022) for the pan-cancer analysis. The gene chips, which included at least 60 samples, were selected from the GEO database; GSE48075 [20] and GSE48276 [21] were used in our research.

### 2.2. Survival Curves, ROC Curves, and Clinicopathologic Features Analysis

On the basis of the EPHA3 expression and survival time data in the GEO gene chips (GSE48075 and GSE48276), the OS curves plotted by the GraphPad Prism software (GraphPad 6.0, Dotmatics, Boston, MA, USA) were applied for evaluating the connection of the EPHA3 level to the BLCA prognosis. Additionally, the prediction ability of EPHA3 expression on the BLCA prognosis was assessed by generating ROC curves and performing a clinicopathologic features analysis, which incorporates clinicopathological data from TCGA-BLCA. The details are listed in Table 1. The ROC results were acquired by using the online analysis website Assistant for Clinical Bioinformatics approach (https://www.aclbi.com/static/index.html#/, accessed on 10 November 2022). The analysis of the clinicopathologic features was determined by the R packages “Limma” [22] and “ggpubr” (Alboukadel Kassambara, (2020)).

### 2.3. Co-Expressed Gene Exploration and Enrichment

Using LinkedOmics (http://www.linkedomics.org/login.php, accessed on 10 November 2022) [23], we downloaded the EPHA3-related genes from TCGA-BLCA. The details of all of the genes are shown in Appendix A. By selecting the top 200 genes, this work conducted the GO and KEGG analysis using Metascape (http://metascape.org, accessed on 10 November 2022) [24]. Then, the results were converted into figures using the Sanger Box tool (http://www.sangerbox.com/, accessed on 10 November 2022). Moreover, this work constructed the protein–protein interaction (PPI) network based on the STRING database (https://cn.string-db.org/, accessed on 10 November 2022) [25], with a basic standard setting of the minimum required interaction score of 0.4. Subsequently, using the networking construction tool, Cytoscape (version 3.9.0, The Cytoscape Consortium, San Diego, CA, USA), we performed the visualization.

### 2.4. Cell Lines and Cell Culture

The present study employed human bladder cancer cells (5637, UMUC-3, and T24), along with normal human uroepithelial cells (SVHUC). Roswell Park Memorial Institute-1640 medium (RPMI-1640; Procell Life Science & Technology, Wuhan, China) was used to culture 5637 and T24 cells, and Dulbecco’s Modified Eagle’s Medium (DMEM; Procell Life Science & Technology, City, US State abbrev. if applicable, Country) was used to culture UMUC-3 and SVHUC cells. The China Center for Type Culture Collection (CCTCC, No: GDC078; Wuhan, China) provided all of the cells. The cell cultures were grown within a cell incubator including 5% CO_2_ under the condition of 37 °C using RPMI-1640 medium and DMEM that contained 1% penicillin-streptomycin (ST551; Beyotime, Wuhan, China), as well as 10% fetal bovine serum (FBS; Ausbian Corporation, Australia).

### 2.5. Cell Transfection

The GV492 lentiviral vectors, which were used to achieve overexpressed EPHA3 levels in the BLCA cells, were provided by Genechem Company (Shanghai, China). Appendix A displays the specific sequences. Then, the vectors were packaged in the HEK293T cells. After transient transfection, the recombinant lentiviruses were produced. Lentiviruses supernatant targeting EPHA3 were used to infect the T24 and UMUC-3 cells, and puromycin was added for 2 weeks to screen the stable EPHA3-overexpressed cells. To assess the transfection efficiency before collecting the cells for the following experiments, an RT-qPCR was performed.

### 2.6. Cell Proliferation Assay

To evaluate the cell proliferation ability, we adopted the CCK-8 assay (C0038; Beyotime). This work inoculated T24/EPHA3, T24/negative control (NC), UMUC-3/EPHA3, and UMUC-3/NC cells (3 × 10^3^ cells/well) in 96-well plates. Afterward, 10 μL reagent was added to all of the wells, and these were left to incubate for 2 h under 37 °C. Moreover, a microplate reader (Enspire 2300 Multimode plate reader; PerkinElmer, Inc., Waltham, MA, USA) was used with the aim of measuring the absorbance in each well at 490 nm. We repeated the same procedure for the next 3 days. Five replicate wells were produced for each sample. When we were performing the statistical analysis, the maximum and minimum values were removed.

### 2.7. Colony Formation Assay

The cells from each group (3 × 10^2^) were seeded in each well and cultivated for 8 days. Four percent paraformaldehyde and crystalline violet (C0121; Beyotime) were used to fix and stained the cells, respectively. The number of colonies with >50 cells in the entire plate was calculated using the microscope.

### 2.8. Cell Migration Assay

The cells (1 × 10^5^) from each group were planted in 6-well plates and cultured at 37 °C until the densities were nearly 100% confluent. Additionally, the wound-healing assay was adopted for assessing the motility of the cells in a clean wound, which was scraped using a 200 µL pipette. Then, the plates were incubated at 37 °C. Photographs of the cell-free area at 0, 24, and 48 h were taken using an inverted microscope (Olympus, Japan). This study performed the calculation of the percentage of wound closure using Image J software (National Institutes of Health, Bethesda, MD, USA).

### 2.9. Cell Invasion Assay

Through the transwell migration assay with the transwell chambers (pore size: 8.0 μm; Corning, NY, USA), the invasion ability of the cells was evaluated. This assay inoculated the cells (5 × 10^5^) in each group into serum-free medium-containing upper chambers. Culture media including 10% FBS were considered an attractant, and they were added into the lower transwell chamber. Following 2 days of incubation under 37 °C, 4% paraformaldehyde and crystalline violet were used to fix and stain the cells that had invaded the lower chamber of the transwell, respectively. After that, the cells were recorded and counted using an inverted microscope (Olympus).

### 2.10. Cell Apoptosis Assay

This assay applied the Annexin V-AbFluor™ 647 Apoptosis Detection kit (KTA0004; Abbkine, Wuhan, China) in the cell apoptosis assay. T24/NC, T24/EPHA3, UMUC-3/NC, and UMUC-3/EPHA3 cells were trypsinized and gathered in 1.5 mL EP tubes. After washing them thrice with pre-chilled PBS, the cells (1 × 10^5^) were selected for resuspension within the 1 × Annexin V Binding Buffer (500 µL), which was followed by staining them with Annexin V-Abflour^TM^ 647 (4 µL) and PI (2 µL) and incubating them in the dark for a quarter of an hour. With the use of flow cytometry (BD Accuri™ C6; Franklin Lakes, NJ, USA), the apoptosis rate of the cells was detected after the incubation.

### 2.11. Animal Experiments

We purchased 4-week-old Balb/C nude mice from Guangdong Yaokang Biotechnology Co., Ltd. (Guangdong, China), which is certified by the Guangdong Provincial Bureau of Science. After adaptive feeding for 1 week, the T24/NC and T24/EPHA3 cells were diluted with PBS to make a 5 × 10^7^ cells/mL suspension. Then, the injection of a 0.1 mL cell suspension was injected subcutaneously into the right forelimb axilla, and the tumors developed after approximately 7 days. After 20 days of tumor formation, each mouse was killed. Meanwhile, the tumor tissue was eliminated. We could calculate the tumor volumes as follows: volume = 0.52*length*width^2^. The approval for the animal study was acquired (see the details in the Ethics approval and consent for participation section).

### 2.12. RNA Isolation and Real-Time Quantitative PCR (RT-qPCR)

One milliliter of TRIzol reagent (15596018; Invitrogen, Waltham, MA, USA) was added to each group for the cell lysis. This work used the NanoDrop 2000 device (Thermo Fisher Scientific, Inc., Waltham, MA, USA) to quantify the total RNA, which was stored in a −80 °C freezer for future use. cDNA was synthesized with the PrimeScript RT kit (RR047A; Takara, Japan).

SYBR Premix Ex TaqII (RR820A; Takara) was adopted to conduct the RT-qPCR using a QS5 PCR device (ABI, Santiago, CA, USA). The gene expression data were normalized against the human reference gene, GAPDH, and the 2^−∆∆Ct^ method was adopted for calculating the result. We detailed the primer sequences in Table 2.

### 2.13. Western Blotting

The T24/EPHA3, T24/NC, UMUC-3/EPHA3, and UMUC-3/NC cells were collected for the protein extraction. The total proteins of the cells were extracted by using a cell lysis solution (P0013B; Beyotime), which included protease (P1006; Beyotime) and phosphatase inhibitors (P1260; Solarbio, Beijing, China). The Bicinchoninic Acid Assay (BCA) kit (P0012; Beyotime) was utilized to assess the protein concentration. The proteins were then heated to 100 °C with sodium dodecyl sulfate-polyacrylamide (SDS-PAGE) protein loading buffer (P0015; Beyotime) for albumin denaturation. After separating using an SDS-PAGE gel, this work electro-transferred the proteins on the nitrocellulose membranes (HATF00010; Merck, Germany). Shortly afterward, we blocked the membranes in 5% Bovine Serum Albumin (BSA) blocking buffer for 1 h. Primary antibodies were used, and they were incubated with the membranes during the night at 4 °C. The details of the antibodies are presented in Table 3. Next, the membranes were thrice washed with TBST, then HRP-labeled goat anti-rabbit secondary antibody (1:5000 dilution, Ab7090; Abcam, MA, US) was added onto the membranes, and they were protected from light for 1 h at an apartment temperature. The ECL Western Blotting Substrate Kit (32209; ThermoFisher) and the chemiluminescence detection system were adopted for detecting the protein signals. The band intensity measurement was performed using the software ImageJ (Version 1.48). The original Western blotting images are listed in Appendix A.

### 2.14. Data Source and Immune Infiltration Expression Analysis

Using the immune infiltration online tool CIBERSORT (https://cibersortx.stanford.edu/, accessed on 10 November 2022), the level of various immune-infiltrating cell enrichments could be assessed. The association between the EPHA3 expression level and the immune-infiltrating cells was determined by the R package “Limma”. For the comprehensive analysis of the relationship between EPHA3 and the immune-infiltrating cells, MCP-counter [26] and TIMER algorithms were used. Moreover, the above techniques were performed based on the Spearman test. Different EPHA3 expression level groups of the TGCA-BLCA samples were employed to calculate the stromal/immune/ESTIMATE scores based on the ESTIMATE [27] algorithm with Wilcoxon rank sum test, and the above results were acquired from the ESTIMATE website (https://bioinformatics.mdanderson.org/estimate/index.html, accessed on 10 November 2022). ICI therapy has better effects on the patients with higher expression levels of the immune checkpoint genes (ICGs) [28]. The connection between EPHA3 expression and 22 common ICGs was explored. We could predict the ICI therapy response through the association between the TIDE scores in each TCGA-BLCA sample and EPHA3 expression. Furthermore, the connection between the EPHA3 level and multiple immune cell gene markers within BLCA was calculated based on the database TIMER (https://cistrome.shinyapps.io/timer/, accessed on 10 November 2022).

### 2.15. Statistical Analysis

GraphPad Prism software (GraphPad 6.0, Dotmatics, Boston, MA, USA) was used for the statistical analyses and to create figures in our article. Three repeated plates were set for each sample, and the sum of three independent trials was acquired to satisfy the statistical analyses. The data are displayed as the mean ± standard deviation from three separate assays. By adopting a paired t-test, pairwise differences between the two groups were assessed. An ANOVA was performed to identify the significant differences between the three groups. In addition, *p* < 0.05 represents statistical significance.

## 3. Results

### 3.1. Flow Charts and Patient Features

The methodologies in our study are listed in a flow chart shown in Figure 1. All of the clinicopathologic features for the TCGA-BLCA cases, containing age, gender, tumor lymph node metastasis (TNM) stage, pathologic stage, histologic grade, subtype, smoking status, lymphovascular invasion status, OS, and disease-free survival (DFS), are listed in Table 3. Based on the median expression rate of EPHA3, low- and high-expression groups were created.

### 3.2. Expression Level of EPHA3 in Normal Samples Was Higher Than in BLCA

As the results are shown in Figure 2, 18 types of cancer including the paired samples were selected to determine the EPHA3 expression level between the tumor and normal tissues. When they were compared with the normal tissues, EPHA3 was strongly upregulated in kidney renal clear cell carcinoma (KIRC, *p* = 8.3 × 10^11^), and lung squamous cell carcinoma (LUSC, *p* = 4.2 × 10^4^). EPHA3 was downregulated in BLCA (*p* = 1.3 × 10^4^), colon adenocarcinoma (COAD, *p* = 1.2 × 10^4^), uterine corpus endometrial carcinoma (UCEC, *p* = 2.1 × 10^3^), kidney chromophobe (KICH, *p* = 2.4 × 10^7^), prostate carcinoma (PRAD, *p* = 0.03), thyroid carcinoma (THCA, *p* = 1.6 × 10^10^), and rectal adenocarcinoma (READ, *p* = 3.9 × 10^3^) when they were compared with the normal tissues.

### 3.3. Higher EPHA3 mRNA Expression Levels May Predict a Better OS Rate in BLCA Patients

We used the GSE database to construct Kaplan–Meier survival curves. The BLCA cases with higher EPHA3 expression levels showed a noticeably better OS rate in the GSE48075 cohort (*p* = 0.027, HR = 0.43 (0.16–1.16); Figure 3A). In the GSE48276 cohort (*p* = 0.0275, HR = 0.4024 (0.1464–1.106); Figure 3B), the same trend of the result was observed.

### 3.4. EPHA3 Was a Good Predictor of Histologic Grade and Status in BLCA

The ROC curves were used for measuring the EPHA3 predictive efficacy in BLCA. When we were predicting the BLCA status (Figure 4A), the AUC was 0.907 (95% CI: 0.849–0.965). The sensitivity was 86.5%, while the specificity was 78.9% when the cut-off value was set at 2.694. When we were predicting the BLCA histologic grade (Figure 4B), the AUC was 0.790 (95% CI: 0.701–0.879). With the cut-off value being determined at 2.694, the sensitivity and specificity were 86.5% and 78.9%, respectively. When we were predicting the BLCA T stage (Figure 4C), the AUC was 0.635 (95% CI: 0.244–1.000). With the cut-off value of 2.834, its sensitivity was 88.8%, while its specificity was 60%. When we were predicting the BLCA M stage (Figure 4D), the AUC of 0.645 (95% CI: 0.527–0.763) was obtained. At a cut-off level of 2.834, its sensitivity was 90%, while its specificity was 42.6%. When we were predicting the BLCA N stage (Figure 4E), an AUC of 0.608 (95% CI: 0.548–0.668) was obtained. At a cut-off level of 1.032, the sensitivity was 50%, while the specificity was 67.4%, separately. According to the above-mentioned results, *EPHA3* could acceptably predict the histologic grade and status within BLCA.

Based on these findings, the EPHA3 mRNA level increased within the normal samples when it was compared with that of the BLCA samples (*p* < 0.001; Figure 5A). Nevertheless, the difference between the EPHA3 expression and the TMN pathologic stages was not significant (*p* > 0.05, Figure 5B–E).

### 3.5. Enrichment Analysis and Interacted Genes Network Construction for EPHA3 in BLCA

As shown in Figure 6A, after setting the false discovery rate (FDR) to < 0.01, there were 6698 co-expressed genes, including 5063 positively- and 1665 negatively-associated genes. The top 50 ones can be found in Figure 6B,C.

Figure 7A shows the GO enrichment results. The enriched biological process (BP) mostly included an extracellular structure organization, an extracellular matrix organization, cell–matrix adhesion, and cell–substrate adhesion. The enriched cellular component (CC) mainly contained an extracellular matrix component, a collagen-containing extracellular matrix, a complex of collagen trimers, and collagen trimers. Additionally, the enriched molecular function (MF) mainly contained an extracellular matrix structural constituent, collagen binding, platelet-derived growth factor binding, as well as an extracellular matrix structural constituent conferring tensile strength. Figure 7B shows the part of the KEGG enrichment results, including the PI3K-Akt pathway, MAPK pathway, vascular smooth muscle contraction, focal adhesion, protein digestion and absorption, dilated cardiomyopathy (DCM), ECM–receptor interaction, cGMP-PKG pathway, calcium pathway, hypertrophic cardiomyopathy (HCM), and the regulation of actin cytoskeleton. Appendix A lists the details of the enrichment results.

As the results show in Figure 8, the PPI network, which was produced using the Cytoscape software (version 3.9.0, The Cytoscape Consortium, San Diego, CA, USA), showed an interaction between the top 200 co-expressed genes and EPHA3, which contained 159 nodes and 491 edges.

### 3.6. EPHA3 Expression within BLCA Cell Lines

According to the findings from the RT-qPCR and WB assays, the EPHA3 expression level was higher in the urinary epithelial normal cell SVHUC than it was in the BLCA cells and the UMUC-3, 5637, and T24 cell lines (Figure 9A–B, *p* < 0.05).

### 3.7. Upregulating EPHA3 Hindered BLCA Cell Growth, Invasion, Migration, and It Enhanced Their Apoptosis

Based on the previous results, we determined that EPHA3 was downregulated in BLCA. To determine the regulating role of EPHA3 in BLCA, a lentiviral vector overexpressing EPHA3 or a control vector with fluorescence was transfected into T24 and UMUC-3 BLCA cells. It was notable that the UMUC-3 and T24 cells expressed lower levels of the internal source of EPHA3. After 24 h of infected cell culturing, the cells stably expressing EPHA3 were screened with puromycin. The RT-qPCR analysis demonstrated a significant increase in the level of EPHA3 expression in the UMUC-3/EPHA3 and T24/EPHA3 cells compared to that in the control cells (Figure 10A). We used these cell lines to ascertain whether EPHA3 affects the proliferation and migration of BLCA cells.

According to Figure 10B,C, the results of the colony formation and CCK8 assays indicated that overexpressing EPHA3 inhibited the proliferation ability of the BLCA cells. Furthermore, the scratch and Transwell assays suggest that BLCA cell migration and invasion can be notably hindered by overexpressed EPHA3 (*p* < 0.05, *p* < 0.001, Figure 10D,E). This work also verified that EPHA3 induced the apoptosis of the BLCA cells (*p* < 0.05, Figure 10F).

### 3.8. EPHA3 Inhibited BLCA Growth In Vivo

Tumor xenograft growth models were constructed using nude mice by the subcutaneous injection of T24/EPHA3 and control cells, then, we observed the xenograft tumor formation on the 7th day. As the result shows in Figure 11, the tumors of the T24/EPHA3 group grew notably slower when they were compared with that of the control group after 20 days (*p* < 0.05), which showed that EPHA3 suppressed the BLCA cell growth in vivo.

### 3.9. EPHA3 Regulated BLCA via the Ras/pERK1/2 Pathway

The Ras/ERK pathway is crucial in regulating BLCA [16,29]. Western blotting was employed to detect the impact of EPHA3 on the Ras/ERK pathway. However, ERK did not present a notable difference between the groups of cells. Based on the results, the upregulation of EPHA3 in both the T24 and UMUC-3 cells inhibited the expression of the Ras protein and phosphorylation of the ERK protein (Figure 12).

### 3.10. EPHA3 Was Correlated with Immune-Infiltrating Cells in BLCA

By adopting the CIBERSORT method, 24 common immune cells were detected with the infiltration enrichment level of BLCA. The correlation with EPHA3 expression is shown in Figure 13A, 22 of which were significantly associated with EPHA3 expression. As the results show in Figure 13B, a variety of immune cells showed a negative association with EPHA3, including the natural killer (NK) CD56 bright cells (*r* = −0.145, *p* = 0.003), Treg cells (*r* = −0.109, *p* = 0.004), and T helper 17 (Th17) cells (*r* = −0.184, *p* = 1.7 × 10^4^). Eighteen immune cell types were favorably related to EPHA3, including the B cells (*r* = 0.342, *p* = 1.11 × 10^12^), anchorage dendritic cells (aDCs) (*r* = 0.098, *p* = 0.046), cytotoxic cells (*r* = 0.113, *p* = 022), CD8+ cells (*r* = 0.142, *p* = 0.004), DCs (*r* = 0.237, *p* = 1.16 × 10^6^), eosinophils (*r* = 0.432, *p* = 1.33 × 10^16^), immature DCs (iDCs) (*r* = 0.349, *p* = 3.6 × 10^13^), mast cells (*r* = 0.516, *p* = 3.72 × 10^24^), macrophages (*r* = 0.444, *p* = 4.56 × 10^17^), neutrophils (*r* = 0.259, *p* = 9.77 × 10^8^), plasmacytoid DCs (pDCs) (*r* = 0.241, *p* = 6.88 × 10^7^), natural killer (NK) cells (*r* = 0.424, *p* = 4 × 10^13^), T cells (*r* = 0.245, *p* = 4.89 × 10^7^), follicular helper T cells [TFHs] (*r* = 0.297, *p* = 8.46 × 10^10^), T effector memory (Tem) cells (*r* = 0.308, *p* = 1.96 × 10^10^), Th1 cells (*r* = 0.306, *p* = 2.59 × 10^10^), T helper (Th) cells (*r* = 0.194, *p* = 7.07 × 10^5^), and Th2 cells (*r* = 0.099, *p* = 0.045). The results of the TIMER algorithm (Figure 13C) indicate that the EPHA3 expression was in positive association with five types of immune-infiltrating cells. The MCP-counter method in Figure 13D indicates that EPHA3 showed a positive relationship with the infiltration of 10 common immune cells. Figure 13E indicates that the higher EPHA3 expression level group was accompanied by increased immune (*p* < 0.001), stromal (*p* < 0.001), as well as ESTIMATE scores (*p* < 0.001) compared with those of the lower EPHA3 expression level group based using the ESTIMATE algorithm.

### 3.11. EPHA3 Was Associated with ICGs in BLCA

Twenty-two common ICGs were analyzed for a correlation with EPHA3 expression (Figure 14A). We showed that 13 ICGs are positively associated with EPHA3, including ADORA2A (*r* = 0.317, *p* = 7.31 × 10^11^), BTLA (*r* = 0.369, *p* = 1.51 × 10^14^), CD244 (*r* = 0.171, *p* = 0.000516), CSF1R (*r* = 0.369, *p* = 1.72 × 10^14^), CTLA4 (*r* = 0.197, *p* = 6.51 × 10^5^), HAVCR2 (*r* = 0.294, *p* = 1.77 × 10^9^), IL10 (*r* = 0.426, *p* < 2.2 × 10^16^), KDR (*r* = 0.446, *p* < 2.2 × 10^16^), LAG3 (*r* = 0.113, *p* = 0.0223), PDCD1 (*r* = 0.1888, *p* = 0.000139), PDCD1LG2 (*r* = 0.264, *p* = 7.01 × 10^18^), TGFBR1 (*r* = 0.255, *p* = 1.92 × 10^7^), and TIGIT (*r* = 0.24, *p* = 1.06 × 10^6^). However, there was no distinction in the TIDE scores between the differential EPHA3 expression groups (*p* > 0.05) (Figure 14B), which indicates that EPHA3 is not related to the response to ICI therapy.

### 3.12. Correlation between EPHA3 and Immune Cells Marker Genes in BLCA

The connection between EPHA3 expression and different immune symbol genes was assessed using the TIMER databases. Eighteen categories of immune cells, containing B, CD8+, and Th1 cells, and their 69 corresponding genes, were selected. The results suggested that EPHA3 was closely correlated with most types of immune gene markers in BLCA (Table 4).

## 4. Discussion

BLCA is a type of common cancer. Approximately 70%–80% of BLCAs are non-muscle infiltrating ones when they are first diagnosed; the others are muscle-invasive tumors, some of which have metastases [30]. The transurethral resection of bladder tumor (TURBT) is the most common therapy in NMIBC, but the probability of tumor relapse is still high, with a 5 year recurrence risk rate of 31–78% [31]. Radical cystectomy, the standard treatment in MIBC, has a 5 year OS rate of only 50% [32]. The early diagnosis of BLCA is beneficial for prompt treatment, and it leads to an improved prognosis [33]. Urine cytology and cystoscopy are still the current gold standards for diagnosing BLCA [34]. However, the existing urine diagnostic markers are not accurate, and cystoscopy is invasive, expensive, and inconvenient, making it unsuitable for routine tests [35,36]. So, it is important to explore new diagnostic markers for BLCA.

EPHA3 was found to be related to cancer occurrence and development, such as lung, colorectal, and hepatocellular cancers [37,38,39]. However, the impact of EPHA3 on BLCA is unknown. In our study, a bioinformatics analysis was used to show that EPHA3 is significantly downregulated in BLCA, and the result was validated using BLCA cell lines. The OS curves indicated that patients with higher EPHA3 expression levels may show a favorable prognosis. Even though the ROC curves indicate that EPAH3 could forecast the status and the histologic grade of BLCA patients. Unfortunately, the bar charts show that the association between EPHA3 and the BLCA clinicopathological data is not significant, and this may be due to the limited number of BLCA cases in the TCGA databases. We will adopt more clinical samples to explore this relationship in the future. To explore the mechanism of EPHA3 in BLCA and its potential effects on BLCA cells, a series of functional studies were performed.

After constructing overexpressed EPHA3 BLCA cell lines, we performed the CCK8 proliferation, plate cloning, cell invasion, and migration, together with apoptosis assays to confirm that EPHA3 inhibits the growth, invasion, and migration of BLCA cells, and it promotes their apoptosis. Furthermore, the in vivo experiments suggest that the upregulation of EPHA3 expression reduces subcutaneous tumorigenesis in the BLCA cells. EPHA3 was shown to be a possible anti-oncogene for BLCA.

In line with the findings of the enrichment results, the Western blotting showed that the activation of EPHA3 may lead to the inhibition of the Ras/pERK1/2 pathway. The RAS protein is a member of the guanosine triphosphate (GTP)-binding proteins, which was the first one to be identified, whose capacity is able to regulate cell growth [40]. Aberrant RAS protein autonomously stimulates cell growth and differentiation, and approximately 20% of tumor cells show mutations in the RAS gene, suggesting that the RAS gene exerts a crucial function in cell growth and differentiation [41]. Overexpressed Ras, together with the mutated CpG island methylation within the promoter region, play important roles in regulating BLCA progression [42]. It has been shown that Ras promotes the phosphorylation of ERK1/2 [43]. H-Ras participates in ERK-mediated pathways to regulate BLCA’s biological behaviors [44,45].

In evaluating the association between EPHA3 and the immune-infiltrating cells, our study suggests that EPHA3 shows a positive relationship with numerous immune cells, including NK cells and CD8+ cells, along with their corresponding markers. Additionally, CD8+ cells are eliminated by tumor-associated macrophages (TAMs), which induce a tolerant tumor microenvironment formation in BLCA [19], and infiltrating levels of TAMs may predict a poor prognosis of NMIBC following Bacille Calmette-Guerin (BCG) infusion therapy [46]. Furthermore, it has been documented that NK cells participate in regulating CD8+ cell-mediated autoimmunity, which provides improved strategies for treating autoimmune diseases and cancers [47]. EPHA3 expression is positively associated with macrophages and cytotoxic cells, which consist of important elements in anti-cancer therapies [48,49]. The analysis of immune cell gene markers indicates that Th17 cell markers, such as STAT3 and IL21R, and IL23R and IL17A, show weak and moderate relationships with EPHA3 expression, respectively. The above results indicate that EPHA3 may participate in regulating the Th17 cells in autoimmunity and cancer. Neutrophils are a special type of immune cells that paradoxically displays anti- and pro-tumor properties [50]. A negative trend exists between EPHA3 expression and neutrophil infiltration, as well as the neutrophil markers (CD15 and CD11b), which indicates that EPHA3 might be correlated with neutrophil activation. Furthermore, the enrichment analysis revealed that EPHA3 and its co-expression genes are closely associated with the PI3K-Akt signaling pathway, which is a classic immune response-related pathway in BLCA [51]. We suppose that EPHA3 targets through the PI3K-Akt signaling pathway to wake up the antigen recognition of the immune-infiltrating cells (such as CD8+ cells) to achieve the purpose of killing the BLCA cells in vivo. These results suggest that the activation of EPHA3 appears to activate immunity in BLCA, thus inhibiting the tumorigenesis and progression of BLCA.

The relationship between EPHA3 expression and 22 common ICIs was analyzed. EPHA3 is positively associated with most of the ICIs, thus activating EPHA3 may help BLCA patients to receive greater benefits from the therapy of ICI, which may lead to a better prognosis. The above results reveal the possible immunotherapy strategy, which involves expressing EPHA3 to enhance the sensitivity of the immune cells to therapeutic drugs.

However, our research still has several limitations, such as a lack of validation in the clinical samples collected by ourselves; an experimental exploration should be carried out to study immune infiltration. We will remedy these shortcomings in our future research.

## 5. Conclusions

To conclude, this work demonstrated that EPHA3 may serve as a biomarker that correlates with a good prognosis, and the histologic grade and status of BLCA. Overexpressed EPHA3 effectively suppresses BLCA cell growth, invasion, migration, and it induces their apoptosis via the Ras/pERK1/2 pathway. EPHA3 is positively associated with various immune-infiltrating cells. EPHA3 is correlated with the infiltration of tumors by the immune cells, which indicates that it may be regarded as a new therapeutic target for BLCA.

## Figures and Tables

**Figure 1 cancers-15-00621-f001:**
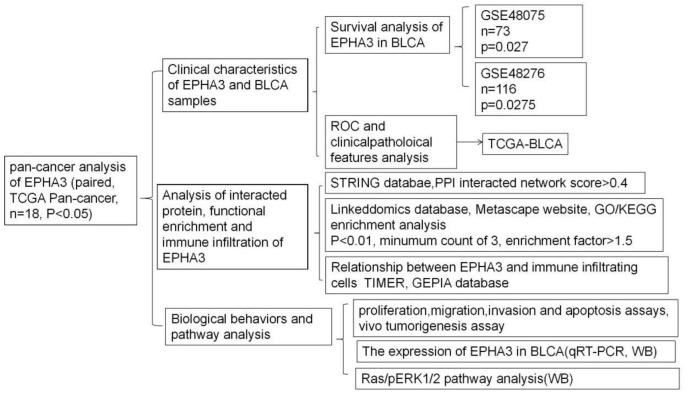
Flowchart showing the research used in this study.

**Figure 2 cancers-15-00621-f002:**
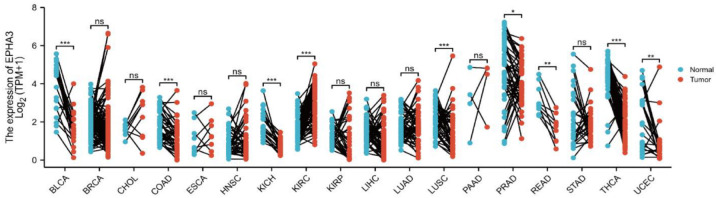
The EPHA3 expression levels in pan-cancers including paired samples in the TCGA database. * *p* < 0.05, ** *p* < 0.01, *** *p* < 0.001, ns stands for no statistical difference.

**Figure 3 cancers-15-00621-f003:**
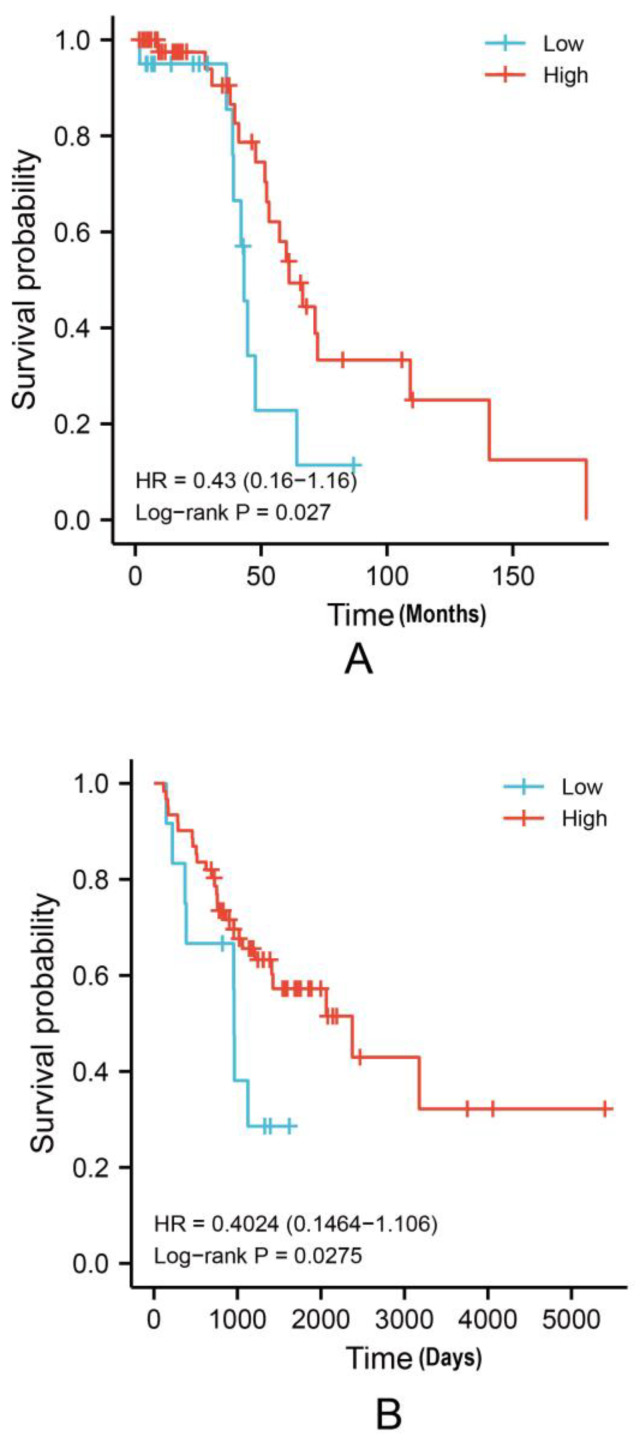
Kaplan–Meier analysis according to low and high *EPHA3* expression levels in GSE48075 (**A**, *p* = 0.027) and GSE48276 (**B**, *p* = 0.0275).

**Figure 4 cancers-15-00621-f004:**
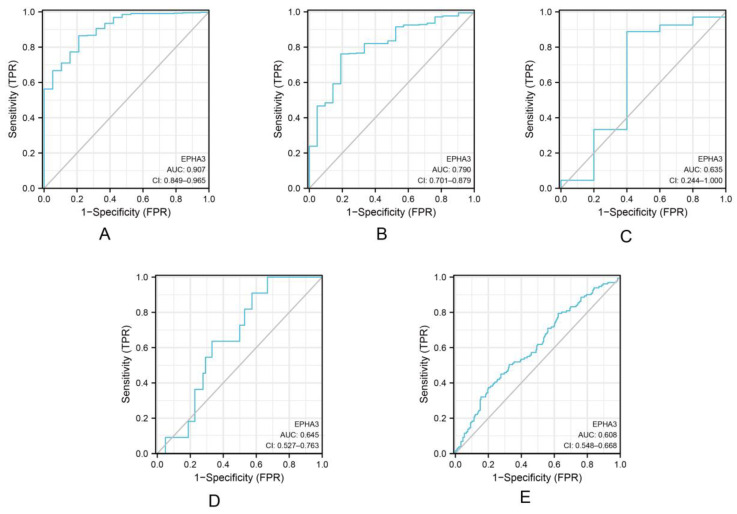
ROC analysis of EPHA3 expression and clinicopathological features from BLCA cases. ROC curves suggest an association between EPHA3 level and status (**A**), histologic grade (**B**), T stages (**C**), M stages (**D**), and N stages (**E**).

**Figure 5 cancers-15-00621-f005:**
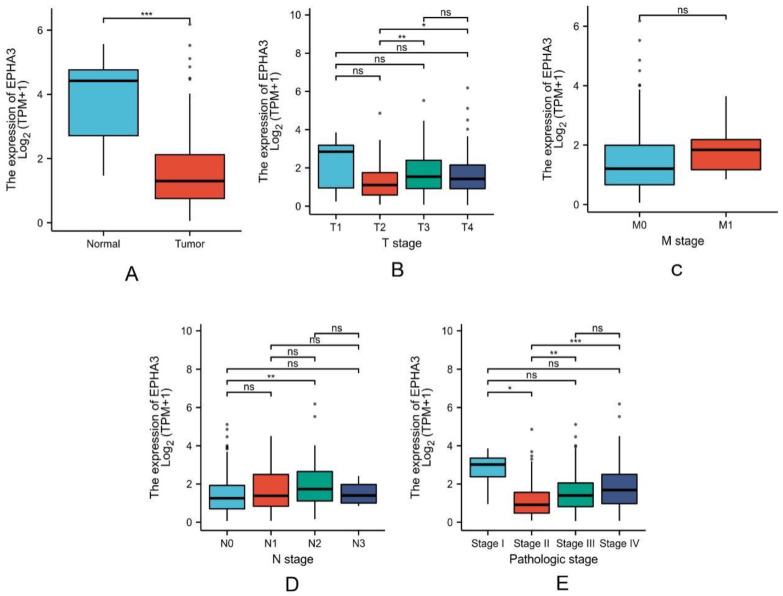
The association between EPHA3 expression and different clinicopathologic data in TCGA-BLCA. The clinicopathologic features included status (**A**), T stage (**B**), M stage (**C**), N stage (**D**), and pathologic stage (**E**). ns stands for no statistical difference, * *p* < 0.05, ** *p* < 0.01, *** *p* < 0.001.

**Figure 6 cancers-15-00621-f006:**
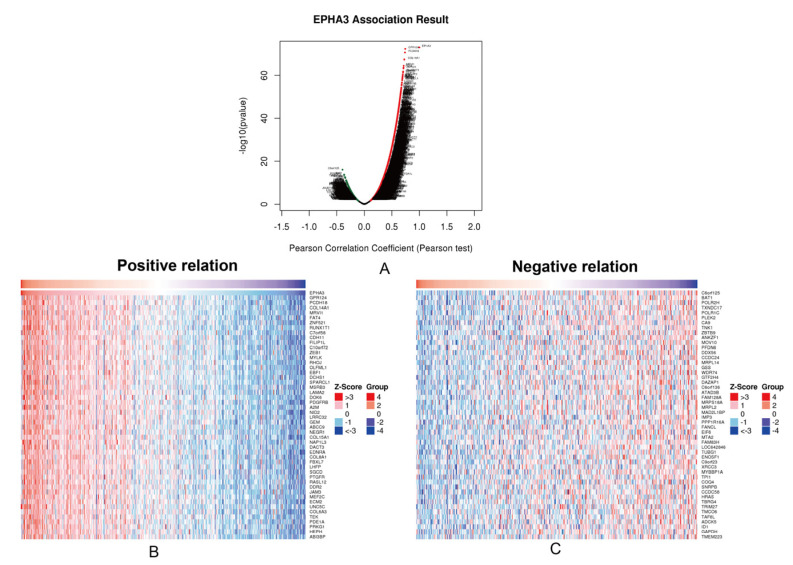
EPHA3 and co-expressed gene analysis within BLCA. (**A**) Volcano plot showing co-expressed genes with a positive and negative relation to EPHA3 within BLCA. (**B**) Fifty most significant genes with positive and (**C**) negative relation to EPHA3.

**Figure 7 cancers-15-00621-f007:**
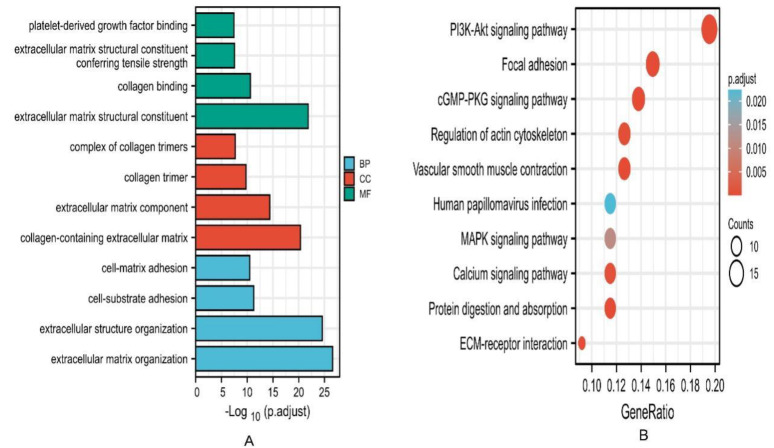
Enrichment analysis of EPHA3 and the most highly related 200 co-expressed genes. (**A**) The findings of GO enrichment analysis in BPs, CCs, and MFs. (**B**) KEGG pathway enrichment analysis.

**Figure 8 cancers-15-00621-f008:**
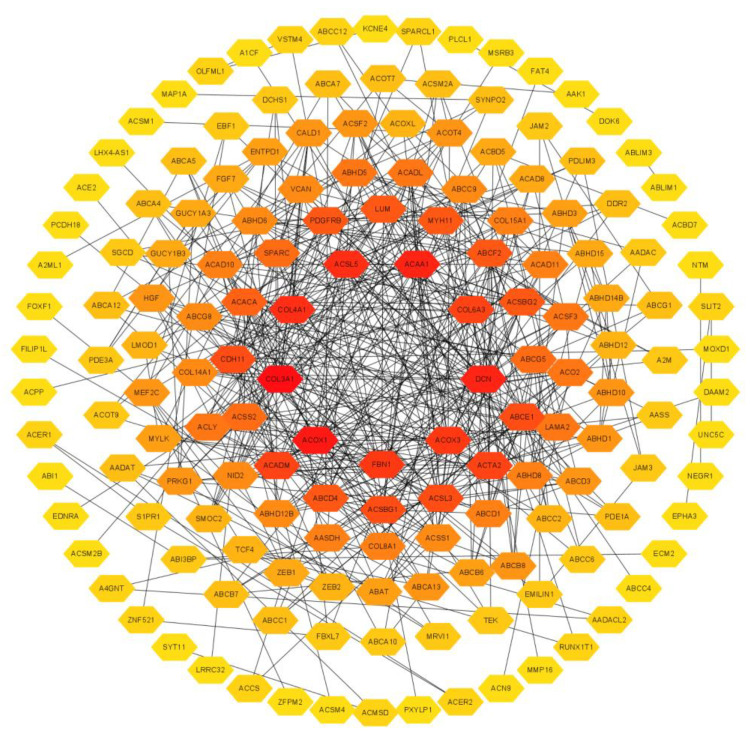
PPI network for EPHA3 and co-expressed genes. The network was composed of 159 nodes and 491 edges, which indicates that EPHA3 may interact with these proteins in BLCA.

**Figure 9 cancers-15-00621-f009:**
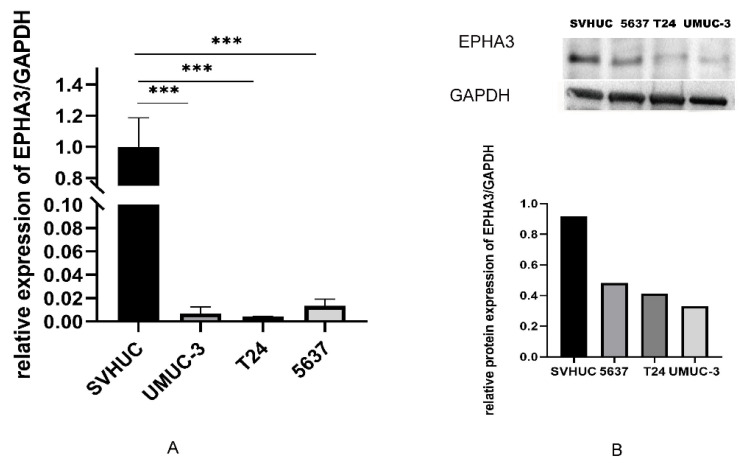
The *EPHA3* levels different cells. (**A**) RT-qPCR on *EPHA3* levels within SVHUC, T24, UMUC-3, and 5637 cells. (**B**) Western blotting of *EPHA3* within the above three cell lines. *** *p* < 0.001.

**Figure 10 cancers-15-00621-f010:**
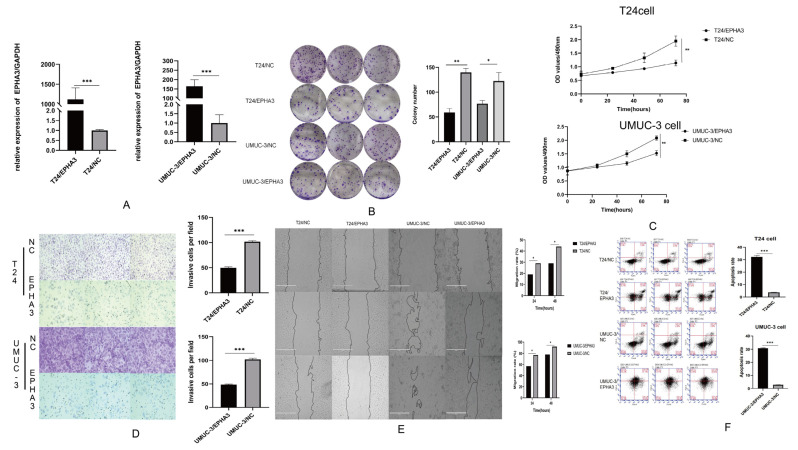
Functional assays of overexpressed EPHA3 and negative control (NC) cells in each group. (**A**) The EPHA3 mRNA levels within T24/NC, T24/EPHA3, UMUC-3/NC, and UMUC-3/EPHA3 groups. (**B**,**C**) The cell proliferation assays were performed by colony formation and CCK8 assay. (**D**,**E**) Changes in cell invasion and migration abilities after overexpressing EPHA3 in T24 and UMUC-3 cells were measured through Transwell and wound-healing assays. (**F**) An enhancement of the number of apoptotic cells was found after overexpressing EPHA3 in T24 and UMUC-3 cells. The total assays were calculated by the software ImageJ (Version 1.48) and the results were visualized by GraphPad Prism software (GraphPad 6.0, Dotmatics, Boston, MA, USA). * *p* < 0.05, ** *p* < 0.01, and *** *p* < 0.001.

**Figure 11 cancers-15-00621-f011:**
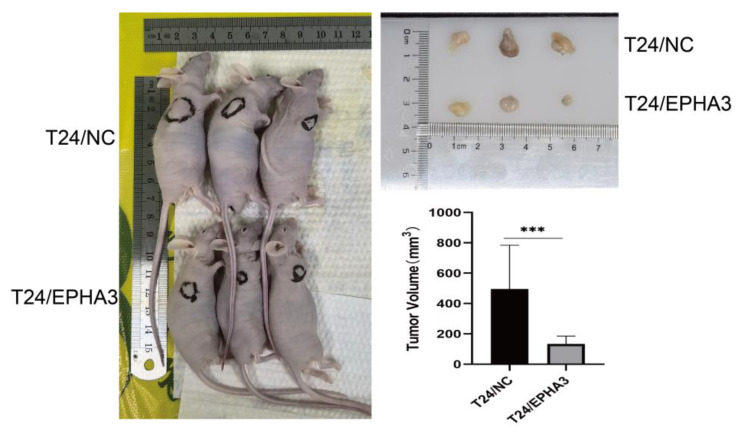
EPHA3 hindered in vivo tumor development. The xenograft tumor volume was measured and recorded. *** *p* < 0.001.

**Figure 12 cancers-15-00621-f012:**
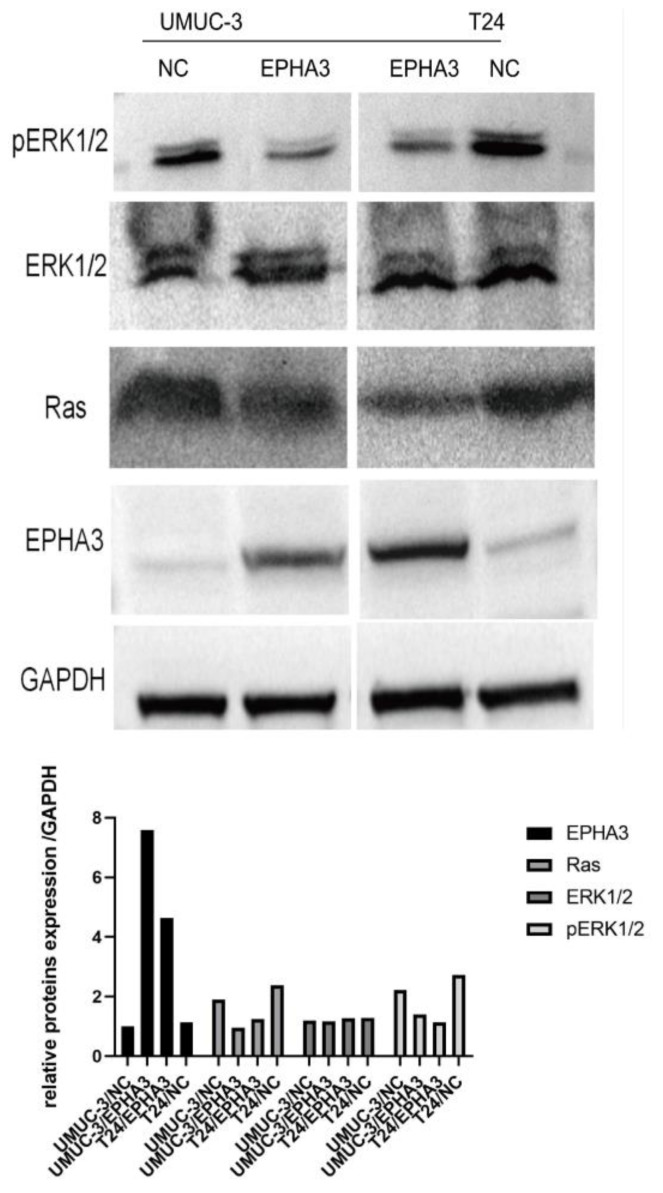
Different protein expressions within T24/NC, T24/EPHA3, UMUC-3/NC, and UMUC-3/EPHA3 groups were quantitatively analyzed by Western blotting. GAPDH was used as a loading control.

**Figure 13 cancers-15-00621-f013:**
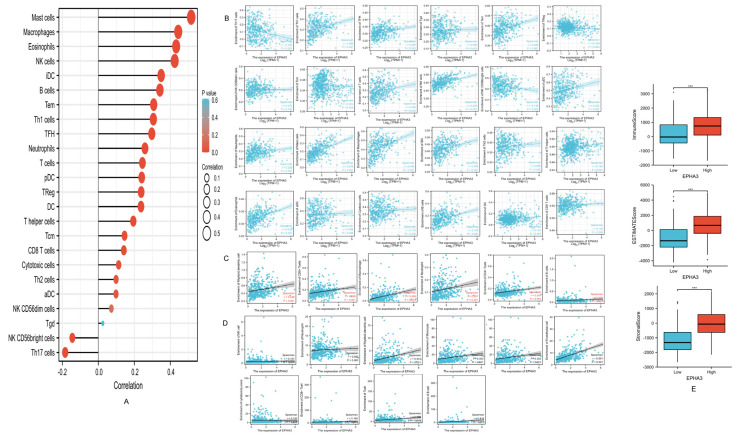
**(A**) Association of EPHA3 level with immune cell infiltration levels in TCGA-BLCA. (**B**) The correlation of EPHA3 level with 24 common immune-infiltrating cells was evaluated by ssGSEA. (**C**) Scatter plots indicating EPHA3 level with 24 common immune cell infiltration levels based on the TIMER method. (**D**) Scatter plots indicating EPHA3 levels with 10 common immune cell infiltration levels using MCP-counter algorithm. (**E**) The stromal, immune, and ESTIMATE scores between the high and low EPHA3 expression groups were explored using the ESTIMATE algorithm. *** *p* < 0.001.

**Figure 14 cancers-15-00621-f014:**
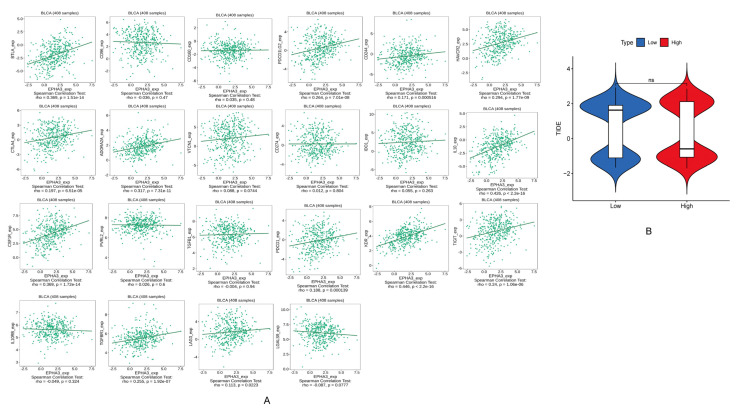
(**A**) Scatter plots indicate the correlation of EPHA3 level with 22 immune checkpoint genes (ICGs). (**B**) The violin plot indicates the association between the TIDE score and high and low EPHA3 expression groups in TCGA-BLCA.

**Table 1 cancers-15-00621-t001:** Clinicopathological features of TCGA-BLCA cases.

Feature	Levels	Low EPHA3 Expression	High EPHA3 Expression	*p*
*n*		207	207	
Gender, *n* (%)	Female	47 (11.4%)	62 (15%)	0.118
	Male	160 (38.6%)	145 (35%)	
Age, *n* (%)	≤70	120 (29%)	114 (27.5%)	0.620
	>70	87 (21%)	93 (22.5%)	
T stage, *n* (%)	T1	2 (0.5%)	3 (0.8%)	0.017
	T2	71 (18.7%)	48 (12.6%)	
	T3	84 (22.1%)	112 (29.5%)	
	T4	25 (6.6%)	35 (9.2%)	
N stage, *n* (%)	N0	122 (33%)	117 (31.6%)	0.225
	N1	21 (5.7%)	25 (6.8%)	
	N2	29 (7.8%)	48 (13%)	
	N3	4 (1.1%)	4 (1.1%)	
M stage, *n* (%)	M0	111 (52.1%)	91 (42.7%)	0.371
	M1	4 (1.9%)	7 (3.3%)	
Pathologic stage, *n* (%)	Stage I	1 (0.2%)	3 (0.7%)	<0.001
	Stage II	84 (20.4%)	46 (11.2%)	
	Stage III	having	77 (18.7%)	
	Stage IV	55 (13.3%)	81 (19.7%)	
Histologic grade, *n* (%)	High Grade	187 (45.5%)	203 (49.4%)	0.002
	Low Grade	18 (4.4%)	3 (0.7%)	
Subtype, *n* (%)	Non-Papillary	122 (29.8%)	153 (37.4%)	0.001
	Papillary	83 (20.3%)	51 (12.5%)	
OS event, *n* (%)	Alive	132 (31.9%)	99 (23.9%)	0.002
	Dead	75 (18.1%)	108 (26.1%)	
DFS event, *n* (%)	Alive	155 (38.8%)	119 (29.8%)	<0.001
	Dead	48 (12%)	78 (19.5%)	
Lymphovascular invasion, *n* (%)	No	75 (26.5%)	55 (19.4%)	0.120
	Yes	73 (25.8%)	80 (28.3%)	
Age, median (IQR)		68 (60, 75)	69 (60.5, 76)	0.379

**Table 2 cancers-15-00621-t002:** Primer sequences adopted in RT-qPCR analyses.

Genes	Sequence (5′–3′)
EPHA3 forward	ATTTTGGCAATGGGCATTTA
EPHA3 reverse	ATGTATGTGGGTCAACATAAGTCC
GAPDH forward	CAGGAGGCATTGCTGATGAT
GAPDH reverse	GAAGGCTGGGGCTCATTT

**Table 3 cancers-15-00621-t003:** Primary antibodies applied in this work.

Primary Antibody	Source Species	Company	Product No.	Predicted Band Size	Dilution
EPHA3	Rabbit	ABCAM	ab126261	110 kDa	1:1000
ERK1/2	Rabbit	CST	#9102	4244 kDa	1:1000
*p*ERK1/2	Rabbit	CST	#4370	4244 kDa	1:2000
Ras	Rabbit	CST	#67648	21 kDa	1:1000
GAPDH	Rabbit	Beyotime	aF1186	36 kDa	1:2000

**Table 4 cancers-15-00621-t004:** The connection between EPHA3 level with immune cell genes markers based on TIMER.

Cell Type	Gene Marker	None		Purity	
		Cor	*p*	Cor	*p*
B cells	CD19	0.361	***	0.253	***
	CD20 (KRT20)	0.032	0.524	0.094	0.072
	CD38	0.307	***	0.16	***
CD8 + cells	CD8A	0.213	***	0.063	0.231
	CD8B	0.175	***	0.066	0.209
Tfh	BCL6	0.074	0.134	0.093	0.0749
	ICOS	0.254	***	0.095	0.0686
	CXCR5	0.371	***	0.253	***
Th1	T-bet (TBX21)	0.248	***	0.084	0.106
	STAT4	0.241	***	0.081	0.119
	IL12RB2	0.107	*	0.008	0.879
	WSX1 (IL27RA)	0.164	***	0.045	0.394
	STAT1	0.133	**	0.014	0.793
	IFN-γ (IFNG)	0.054	0.277	−0.089	0.087
	TNF-α (TNF)	0.084	0.0913	−0.028	0.599
Th2	GATA3	−0.058	0.24	0.029	0.585
	CCR3	0.158	***	0.122	*
	STAT6	−0.023	0.637	0.031	0.557
	STAT5A	0.49	***	0.42	***
Th9	TGFBR2	−0.014	0.902	−0.022	0.855
	IRF4	0.358	***	0.215	***
	PU.1 (SPI1)	0.384	***	0.26	***
Th17	STAT3	0.244	***	0.164	**
	IL21R	0.346	***	0.215	***
	IL23R	0.168	***	0.127	*
	IL17A	−0.084	0.094	−0.136	**
Th22	CCR10	0.226	***	0.211	***
	AHR	0.067	0.177	0.139	**
Tregs	FOXP3	0.409	***	0.319	***
	CD25 (IL2RA)	0.364	***	0.227	***
	CCR8	0.441	***	0.363	***
T cells exhaustion	PD-1 (PDCD1)	0.234	***	0.071	0.176
	CTLA4	0.237	***	0.079	0.129
	LAG3	0.171	***	0.014	0.718
	TIM-3 (HAVCR2)	0.345	***	0.206	***
Macrophages	CD68	0.177	***	0.06	0.25
	CD11b (ITGAM)	0.376	***	0.255	***
M1	INOS (NOS2)	0.225	***	0.182	***
	IRF5	0.072	0.141	0.074	0.155
	COX2 (PTGS2)	0.172	***	0.122	*
M2	CD163	0.42	***	0.314	***
	ARG1	0.136	**	0.164	**
	MRC1	0.404	***	0.304	***
	MS4A4A	0.453	***	0.356	***
TAMs	CCL2	0.445	***	0.329	***
	CD80	0.257	***	0.121	*
	CD86	0.355	***	0.229	***
	CCR5	0.323	***	0.15	**
Monocytes	CD14	0.333	***	0.192	***
	CD16(FCGR3B)	0.193	***	0.109	**
	CD115 (CSF1R)	0.415	***	0.312	***
Neutrophils	CD66b (CEACAM8)	0.047	0.341	0.032	0.539
	CD15(FUT4)	0.284	***	0.184	***
	CD11b (ITGAM)	0.376	***	0.255	***
NK cells	XCL1	0.055	0.264	0.075	0.15
	CD7	0.208	***	0.034	0.501
	KIR3DL1	0.128	**	0.044	0.397
DCs	CD1C(BDCA-1)	0.324	***	0.242	***
	CD141(THBD)	0.093	0.0601	0.037	0.479
	CD11c (ITGAX)	0.406	***	0.285	***

* *p* < 0.05; ** *p* < 0.01; *** *p* < 0.001.

## Data Availability

All data adopted in the current work can be obtained from the TCGA (https://portal.gdc.cancer.gov/) and GEO databases (http://www.ncbi.nlm.nih.gov/geo/). In addition, all authors have no particular access privileges.

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
