# Peer review of "EPHA3 Could Be a Novel Prognosis Biomarker and Correlates with Immune Infiltrates in Bladder Cancer"

_cancers, 2023, doi:10.3390/cancers15030621_

Round 1

Reviewer 1 Report

1. Many grammatical errors

2. Figures are not prepared in professional way: low quality, low resolution (I can't follow up the results since I can't read any texts shown in figures), Samples in western blot should be ordered in the same way as you showed in the graph, the original blots should also be labeled, Fig. A11 should be clearly labeled.

3. Table 1: The clinical data shown in the table should start from Gender and Age followed by stages and grading

4. In the introduction, the authors should mention about the evidence to link between EPHA3 and immune infiltration or immunotherapy.

5. Result 3.2: authors should clarify about the results whether EPHA3 is upregulated or downregulated in cancers, and why EPHA3 in BLCA is interesting.

6. Result 3.4: What is BLCA status?

7. From the results, it is not convincing enough to conclude that EPHA3 could be a novel biomarker for prognosis. The authors should clearly explain/suggest how we can apply EPHA3.

8. Also, it is not convincing about EPHA3 and immune infiltration. I expected to see some gene ontology related to immune response (from enrichment analysis).

9. In general, the authors should interpret and discuss the results more extensively to convince your results.

10. All formats should be consistent; 1h or 1 h, qRT-PCR or QPCR etc.

11. Ethical concern: the approval of animal study should be mentioned.

Reviewer 2 Report

I found the manuscript and the study very interesting and accurate.

-Background and methods are clear. However, I would better clarify the main endpoint (“Based on the above exploration, the exploration aimed to grope for the molecular 77 mechanism of EPHA3 in BLCA.”) this is not a very clear phrare and the word “exploration” is redundant.

- It’s not easy to understand how many researchers were involved in the dataset analysis and evaluation

- Results and tables / figures are well presented

-Limitations of the study are missed and should be implemented.

Reviewer 3 Report

Summary:

This study proposed that EPHA3 could be a biomarker of BLCA, and found that EPHA3 is correlated to many of immune infiltrating cells.

Major comments:

1. The authors mentioned in the conclusion "this work demonstrated that EPHA3 may serve as a biomarker in predicting good prognosis, histologic stage and status in BLCA." However, the authors also mentioned in the 3.2 that some other cancers are also highly correlated to EPHA3. Then, how can we know the EPHA3 expression is not caused by the other cancers? 

2. In 3.4, the authors claimed that EPHA3 was a good predictor of pathologic stage and status in BLCA. However, it seems that it is not a good predictor from the plots. How did the authors decide that it is a good predictor? Please provide some references to show that a gene with such data is considered as a good predictor of a cancer.

3. Please give more elaboration of Figure A8. It is meaningless to put a network plot without clear explanation.

Minor comments:

1. Seems a typo in line 280. Should be "EPHA3" not "EPHA"?

2. I suggest to use square brackets for citation. Round brackets may cause some misunderstanding.

Reviewer 4 Report

The paper of Liu et al is devoted to study the significance of EPHA3 in bladder cancer. The authors have shown its prognostic value and onco-suppressive properties.

For my opinion, several issues should be addressed:

1)    Figures # 10, 13 and 14 are absolutely unreadable (low quality)

2)    29…30 “Overexpressed EPHA3 could inhibit BLCA cell biological behaviors through the Ras” pERK1/2 pathway”

The authors cannot confirm this, they did not use any inhibitors or Ras knock-down to prove it. They only can say that the overexpression of EPHA3 is associated with down-regulation of Ras-ERK1/2 pathway.

3)    To better study the biological significance of EPHA3 in bladder cancer, the authors should carry out knock-down of this gene, for instance, in SVHUC cell line which expresses it at the high level (Figure A9) followed with colony-formation, proliferation and migration assays.

4)    483…484    “EPHA3 promotes tumor immunity..”.

Did the authors carry out such experiments on the impact of EPHA3 on tumor immunity? On my opinion, we can only speak about the association between the level of EPHA3 and the infiltration of tumors by immune cells.

5)    483…484   “EPHA3 promotes tumor immunity, which indicated that it may be regarded as the new therapeutic target in BLCA” and 32…33 “EPHA3 plays an inhibiting role in BLCA, it could be the candidate immunotherapeutic target for BLCA”.

What do authors mean by the word “therapeutic target”? They concluded that EPHA3 could be a marker of favorable prognosis. Do authors imply some drugs that can stabilize or increase the level of this protein?

Round 2

Reviewer 1 Report

Dear authors,

Thank you for your revision of this manuscript. However, here are my comments on the writing and experiments:

1.  Please check the consistency of these formats and change them into the same formats:

a.       CD8+ T cell should be CD8+

b.       Cell line: UMUC-3 or UMUC3

c.       X hours, xh, x h

d.       Cell number: 3*102 or 3X102

2. If NC is negative control, the authors should give the definition in the figure description. Also, the definition of OS (Overall survival rate) should be mentioned (for example line 93, 258).

3.       Western blot analysis, which instrument did you use for detection and which software did you use for band intensity measurement?

4.       Line 176: 5×107/ml suspension. Should be cells/mL?

5.       Labelling of the plots and graphs should be clear:

a.       Figure 3: number of specific cohorts should be clearly shown in the graph. And what is the time in X-axis (Months?)?

b.       Figure 6: B = Positive relation, C = Negative relation. Should be labeled under the graphs

c.       As well as figure 7 and 9, please label figure A and B

d.       Figure 11: From mice and tumor photos, which group is negative control and which group is EPHA3 overexpression? These should be labeled in the photos.

6.       I noticed that you have changed the figure number (for example, from A1 to 1), please check the number of your figures in the paragraphs as well.

7.       Line 248: EPHA3 was strongly denoted in kidney renal clear 248 cell carcinoma (KIRC, P=8.3e-11), colon adenocarcinoma (COAD, P=1.2e-04), uterine cor- 249 pus endometrial carcinoma (UCEC, P=2.1e-03), and lung squamous cell carcinoma (LUSC, 250 P=4.2e-04). However, I would suggest to explain this results into which ones are upregulated or downregulated.

8.       Line 319: What is SLIT2 and what is its significance to this result?

9.       Figure 10:

a.       The order of samples/conditions in every graphs should be the same so the readers can follow the results easily.

b.       Graph 10B should be bigger

c.       Label the assay under or above the graphs, for example Transwell assay in figure 10D.

d.       10E: cannot clearly see the gap closure, I would suggest drawing the line to indicate the gap area. And the graphs are difficult to read.

10.   Figure 12: The order of samples/conditions in WB should be the same between different cell lines and the band intensity should be presented in bar graph.

11.   Result 3.4: the explanation is confusing between stage and grading. As I understand, EPHA3 should be a biomarker for histological grading (PUNLMP/Low grade/High grade). If this is correct, the authors should clarify about this. Stages are more related to Stage I/II/III… or metastasis.

12.   And if it is related to histological grading, do you have the expression levels between PUNLMP/Low grade/High grade? (Compare to figure 5)

13.   In conclusion: Overexpressed EPHA3 effectively suppresses BLCA cell growth, invasion, migration and induces their apoptosis 502 via the Ras/pERK1/2 pathway. From the results, it is not convinced yet. I encourage the authors to add the experiment to measure the expression of downstream genes in Ras/pERK1/2 pathway that related to cell growth, proliferation or apoptosis after EPHA3 overexpression.

14.   Knockdown of EPHA3 in normal cell (SVHUC) would be beneficial to confirm the molecular mechanism of this gene too (if possible).

Reviewer 3 Report

1. I understood that the authors used BLCA specific dataset to find out that EPHA3 may serve as a biomarker of BLCA. However, if there is a "new" patient, and he / she has BLCA and maybe other cancers or not, then how can one know the expression of EPHA3 is caused by what? The point is, EPHA3 is not a BLCA-specific biomarker, so I doubt that it is really useful for predicting anything, since it could be affected by many other cancers.

Reviewer 4 Report

The authors have addressed main points which I have asked them. Thus, I suggest to accept the paper in the present form

Author Response

Thank you for your praise, I wish my manuscript can be accepted soon.

Round 3

Reviewer 1 Report

1.       Line 250: EPHA3 was strongly upregulated in kidney renal 250 clear cell carcinoma (KIRC, P=8.3e-11), colon adenocarcinoma (COAD, P=1.2e-04), uterine corpus endometrial carcinoma (UCEC, P=2.1e-03), and lung squamous cell carcinoma (LUSC, P=4.2e-04). However, from Fig.2, only KIRC obviously showed the upregulation of EPHA3. COAD, UCEC and LUSC tend to show downregulation. Can you explain why you conclude that EPHA3 is upregulated in these 3 cancers?

2.       The reviewers have asked if the authors performed EPHA3 knockdown experiment in SVHUC cells or not. Partly, the reasons are to confirm if EPHA3 regulates the genes through Ras pathway or not, or after EPHA3 downregulation in normal cells, is there a possibility that this event induce the transformation of normal cells into cancer cells or not.

Reviewer 3 Report

I have no further comments.